# Construed Organizational Ethical Climate and Whistleblowing Behavior: The Moderated Mediation Effect of Person–Organization Value Congruence and Ethical Leader Behavior

**DOI:** 10.3390/bs14040293

**Published:** 2024-04-01

**Authors:** Han Cai, Lingfeng Zhu, Xiu Jin

**Affiliations:** Department of Business Administration, Gachon University, Seongnam-si 13120, Gyeonggi-do, Republic of Korea; ch11245212@163.com

**Keywords:** organizational ethical climate, organizational identification, person–organization value congruence, leader ethical behavior, whistleblowing behavior

## Abstract

An organizational ethical climate enhances the degree of collaboration and cohesion among employees and facilitates the development and interests of organizations. Such roles lead to organizational sustainable development and survival. Therefore, the importance of ethical climate in organizations is becoming increasingly apparent. In this background, this study aims to explore whether an organizational ethical climate can improve whistleblowing behavior and the mediating role of organizational identification in promoting whistleblowing behavior. Most previous studies have only focused on the mediating or moderating role of the model. This study expands the research field, adds the dual moderation of person–organization value congruence and leader ethical behavior, and verifies two moderated mediation models. Overall, the purpose of this study is to determine the behavior of employees under the influence of an organizational ethical climate and, on this basis, propose suggestions for strengthening organizational ethical climate, expanding the scope of research on organizational climate and providing a theoretical basis for related research. In order to achieve the research goals, the data were collected from 344 Chinese SMEs for empirical analysis. The results showed that an organizational ethical climate has no direct impact on whistleblowing behavior but could have a positive effect on whistleblowing formation through the mediating variable of organizational identification. In addition, person–organization value congruence and leader ethical behavior significantly moderated the mediating role of organizational identification between organizational ethical climate and whistleblowing behavior. Finally, the directions that can contribute to future research were suggested.

## 1. Introduction

In recent years, people have paid increasing attention to the phenomenon of unethical behavior in organizations to safeguard their interests. With frequent organizational and work ethics crises, moral issues transcend the scope of traditional philosophical research and have gradually become a hot spot of widespread concern in theoretical and practical circles of organizational management [1]. In the context of social governance innovation, various social organizations in my country that undertake public welfare, professional social responsibilities, and public service functions transferred by the government have expanded in scale and developed rapidly [2]. An organizational ethical climate, as an organizational climate characterized by pro-environment practices and morality, can promote the altruistic behavior of organizational employees and formulate clearer ethical standards [3]. An organization’s ethical climate refers to the extent to which individuals, as members of an organization, perceive the ethics of organizational procedures, policies, and behaviors [4]. An ethical climate emerges when organization members believe ethical standards or norms for actions or decisions are desirable within the organization [5]. When employees feel a strong organizational ethical climate, they increase their trust in the organization’s goals and improve their emotional attachment to the organization [6]. Therefore, these highlights indicate the importance of an ethical climate in organizations, which should strengthen ethics to create a positive working atmosphere for employees.

In addition, according to social information processing theory, employees’ attitudes and behaviors depend on the information available in their social environment. Therefore, an organizational ethical climate, as a universal perception and experience for employees, will affect their psychology and behavior [7]. An ethical climate influences employees’ ethical judgment as they assess behavior based on whether it harms others or the organization, violates ethical standards, or complies with laws, which is crucial for whistleblowing intentions [8]. Therefore, to a certain extent, employees’ whistleblowing behavior can help leaders manage the organization more effectively and suppress incivility within the organization [9]. When employees feel the organization has an ethical climate, they tend to prevent unethical behavior by reporting to aid the long-term development of the organization. Therefore, this study predicts that an organizational ethical climate can increase employees’ whistleblowing behavior.

This study aims to determine whether an ethical organizational climate can enhance employees’ whistleblowing behavior. Organizational identification is believed to have a mediating effect between an organization’s ethical climate and employee whistleblowing behavior. The level of an organization’s ethical climate plays an important role in its organizational identification. According to social identity theory, if an organization focuses on ethical behaviors, most group members will copy these behaviors [10]. When employees with a high degree of organizational identification are watching or discovering unethical behavior, they will take an organizational perspective to avoid it by reporting it and seeing the act of whistleblowing as a way to protect the organization [11]. Therefore, employees who work in an ethical climate can feel the values of the organization and identify more with them. They view things from the perspective of organizational ethics and stop unethical behavior through reporting. Therefore, an ethical organizational climate has a positive impact on whistleblowing behavior through organizational identification.

In addition, this study argues that employees’ whistleblowing behaviors change in response to moderated mediation of person–organization value congruence. Person–organization value congruence refers to the degree of fit between organizational and employees’ personal values [12]. Individual behavior is determined and guided by the work environment and individual decisions are constrained by internal characteristics and the external environment. When the external environment conforms to the individual’s internal characteristics, the individual will regard work as a form of self-expression, which will lead to positive work attitudes and behaviors [13]. Once the match between the individual and the organization is high, people feel more comfortable in the organization [14]. Therefore, this study demonstrates that person–organization value congruence strengthens organizational identification, thereby providing reasons for employees’ whistleblowing behavior. Therefore, exploring the moderating effects of person–organization value congruence is necessary. The interaction between ethical organizational climate and person–organization value congruence increases the occurrence of employee whistleblowing.

This study argues that changes in employees’ whistleblowing behavior also change with the moderated mediating effect of leaders’ ethical behavior. Ethical leaders have moral qualities of honesty, integrity, justice, fairness, and care for others. This allows employees to believe they can point out problems and flaws in the organization without suffering retaliation from the leader, even if they publicly point out the leader’s mistakes [15]. Therefore, this study assumes that leaders’ ethical behavior strengthens organizational identification, thereby providing reasons for whistleblowing. Therefore, exploring the moderating effect of ethical leaders’ behavior is necessary. The interaction between organizational ethical climate and leaders’ ethical behavior will increase employee whistleblowing.

Based on the above theory, the purpose of this study is as follows. First, the complex and changing business environment makes employees have higher requirements for the organizational climate. When employees feel the ethical behaviors and values of the organization, it increases their psychological contract with the organization and creates empathy, which contributes to the increase in employees’ psychological security, reduces negative expectations of whistleblowing behaviors, and increases their likelihood of taking action [7]. In addition, by collating previous findings, we confirmed that there is a relative lack of empirical research on organizational ethical climate and employee whistleblowing in our country. Therefore, this study elucidates the relationship between an ethical organizational climate and whistleblowing behavior. It demonstrates how an ethical organizational climate leads to employee whistleblowing, which will help expand the field of research on organizational ethical climate.

Second, most studies explore the outcome variable of the organizational ethical climate [16,17,18]). The mediating or moderating effect of an organization’s ethical climate on the induction process [19,20,21]) has been verified. However, we expanded the research climate to include the organization’s ethical climate. Furthermore, we propose and validate a moderated mediation model.

Third, this study identifies and examines the moderating effects of person–organization value congruence and ethical leadership behavior. Specifically, by presenting the interactions between organizational ethical climate, person–organization value congruence, and leader ethical behavior, we have determined how the interaction effects altered whistleblowing behavior and moderated the mediating effects of organizational identification.

Finally, we lack research on the organizational climate in Chinese SMEs and elucidate and examine the role of organizational ethical climate in Chinese SMEs. This will help expand the field of research on ethical organizational climates. Specifically, this study proposes a new research model and reveals how an organization’s ethical climate affects the process of employee whistleblowing. This study contributes to expanding the research field of ethical organizational climate and whistleblowing behavior. In addition, it reveals the extent of the organizational climate in Chinese SMEs and helps explain the role of person–organization value congruence and ethical leadership behaviors in Chinese SMEs through the interplay of organizational identity.

## 2. Theoretical Background and Hypotheses

### 2.1. The Mediating Effect of Organizational Identification

An organization’s ethical climate is the unanimous perception of its members about the characteristics of its ethical environment [22]. In organizations, when ethical standards of behavior are maintained and an ethical atmosphere is formed, employees can benefit from cultivating clear moral cognition [23]. An ethical climate refers to an organization’s shared views on formal and informal policies, practices, and procedures [24]. From the perspective of social communication, a strong ethical climate allows employees to feel that the organization cares about their interests, which enhances their psychological attachment and organizational commitment [8]. Research on organizational ethical climate has found that it is positively related to organizational identification [25] and commitment [6]. Based on these theories, this study argues that an organization’s ethical climate refers to the organization requiring its members to conform to ethical principles and values, to respect the power of the employees, and to safeguard the interests of the employees and it is an experience and perception of the employees about the organizational climate.

Organizational identification refers to the psychological connection between an individual and the organization; that is, a deep self-defining emotional and cognitive connection between the individual and organization [25]. Employees who highly identify with the organization will have a positive attitude toward stopping wrongdoing within the organization [26]. Organizational identity refers to the psychological connection between employees and their organizations and their emotional value [23]. Employees with strong organizational identification act beyond their own interests for the overall good of the organization [27]. Research has found that organizational identification is positively related to employees’ responsible behavior [28] and organizational satisfaction [29]. Based on these theories, this study argues that organizational identification refers to the consistency between individual employees’ perceptions of many concepts and the organization’s practices. Therefore, when employees have a high level of organizational identification, they feel responsible for the organization and are more willing to devote themselves to it for the sake of its interests and goals.

Whistleblowing behavior is reporting of unethical behavior by an organization to a higher authority for correction [30]. Whistleblowing behavior in an organization safeguards the long-term interests of the organization, helps promote the development of individuals and organizations, and stimulates positive cognitive evaluations by bystanders [1]. It also refers to employees reporting inappropriate behavior in an organization [31]. Moreover, whistleblowing behavior elicits a positive work response in organizations and benefits both the organization and its employees in reporting and correcting misconduct [32]. Research on whistleblowing behavior has found that organizational identification is positively related to employees’ internal whistleblowing behavior [26]. Whistleblowing and job performance show a positive correlation [33]. Based on these theories, this study suggests that whistleblowing behavior refers to employees’ whistleblowing to internal or external agencies of the organization to expose behaviors that undermine the interests of the organization or violate laws. Whistleblowing behavior in an organization can improve its ethical standards, reduce losses, and positively affect the interests of the organization.

An ethical climate shows the justice of management and demonstrates that an organization values the contribution of its employees and that this can enhance employees’ trust in the organization and willingness to attach to the organizational relationship, thus promoting their organizational identification [8]. When leaders create an ethical climate in their organization and give it to their employees through empowerment, perceived as supportive of leadership and based on collective contributions, employees feel more connected to their organization and thus organizational identification is created [23]. Accordingly, when employees share the same perception of the company’s ethical climate, they are likely to have a clear understanding of the ethical behaviors expected from their organization, thus creating organizational identification [24]. Moreover, employee whistleblowing behavior is based on an effective whistleblowing system; when policies within the organization encourage employees to whistleblow, the ethical climate of the organization is altered and enhanced, which, in turn, leads to whistleblowing behavior that reduces the occurrence of errors in the organization [34]. In an ethical organizational climate, employees strictly adhere to the organization’s policies and use them to make ethical decisions and solve problems, which, in turn, gives rise to whistleblowing behavior among employees [35]. As well employees with a high level of organizational identification being more inclined to act in the interests of the organization rather than in their own, whistleblowing behavior can be used to detect problems in the organization in time for early warning [8]. The more satisfied employees are with their current organization, the more they want to have a voice in the organization and whistleblowing behavior for any violation in the organization satisfies their needs and expectations [36].

Specifically, in organizations, when employees perceive an ethical climate, it promotes employee organizational identification, which increases whistleblowing behavior [37]. An ethical climate in an organization stimulates employees’ proactive personalities, strengthens their organizational identification, and leads to whistleblowing behavior [26]. Moreover, ethical leadership behavior promotes ethical behavior through the implementation of its principles and creates a positive ethical climate in the organization. This helps employees adapt well to the ethical standards of the organization, which results in a high level of organizational identification. This encourages employees to use whistleblowing to maintain the organization’s ethical norms and report unethical behavior [10]. In addition, in an organization with an ethical climate, employees define themselves by the organization’s ethical standards and highly ethically identified employees engage in whistleblowing in favor of the organization’s interests and goals, regardless of how high the risk of disadvantage is [8]. When an organization has a strong ethical climate, it shares ethical issues with its employees in the process of institutionalizing its code of ethics, which leads employees to believe that their organization takes their ethical issues seriously. This increases their organizational identification and makes them more willing to engage in whistleblowing behavior for the benefit of the organization and card sustainability [38]. Based on these theories, this study proposes the following hypotheses:

**Hypothesis** **1.**
*Organizational identification will mediate the relationship between organizational ethical climate and whistleblowing behavior.*


### 2.2. The Moderated Mediation Effect of Person–Organization Value Congruence

Person–organization value congruence is the extent to which personal values are similar to those of the organization [39]. Person–organization value congruence applies when organizational core values match individual values well [40]. Thus, it may be important across a wide range of jobs and careers [41]. Moreover, with a high degree of per-son-organization value congruence, employees perceive that they are already a part of the organization’s life and that the organization is part of their identity [42]. Person–organization value congruence is defined as the congruence between the norms and values of the organization and those of its people [43]. In organizations, high-value congruence represents good compatibility between the leader and the organization, which can reduce employee burnout [42]. Research on person–organization value congruence has found that it is positively related to job satisfaction [41] and organizational performance [39]. Based on these theories, this study considers person–organization value congruence to be the degree of match between an employee’s own values and the organization’s values. When a high degree of value congruence exists between the employee and organization, the employee treats the organization’s goals as objectives and generates a work ethic to help the organization’s sustainable development.

This study emphasizes the moderating role of person–organization value congruence and argues that this variable increases the role of organizational ethical climate in organizational identification. Therefore, the organizational identification of employees in Chinese SMEs is determined by the interaction between the organization’s ethical climate and person–organization value congruence. An employee with a high degree of organizational value congruence will have a higher level of identification with the organization, which will lead to more dedication, satisfaction, and effort toward the organization’s successful cause [44]. It can be effective in promoting employee engagement in behaviors that are beneficial to the organization at work, thus positively affecting it [45]. Organizational identification can be seen as a result of value congruence, so when employees are more open and communicative in the workplace, they can better understand the organization and participate in organizational communication, which leads to an increase in value congruence with the organization. When employees’ person–organization value congruence is high, it their trust and commitment to the organization increases, which boosts their organizational identification [46]. Person–organizational values congruence refers to the congruence between the individual characteristics of the employee and the organization, in which the employee realizes his or her own self-worth in the goals and objectives of the organization, thus showing a positive attitude toward the organization and increasing organizational identification [47]. Particularly in Chinese SMEs, when the values of employees are aligned with the organization, they have a positive impact on the organization and this positive impact increases employees’ organizational identification. Therefore, when employees’ values are strongly matched with those of the organization, they accept the goals of the organization, are more willing to make efforts toward those goals, and show strong organizational identification, which will contribute to the development of organizational goals [47]. Therefore, according to these theories, when employees have a high degree of value congruence with their own organization, they feel more deeply that they have become part of the organization’s life, thus increasing overall positive feelings toward the organization and contributing to an increase in organizational identification.

In addition, an ethical climate can make employees feel that their ethical values are aligned with those of the organization and motivate them to be more loyal to and willing to stay in the organization [48]. Therefore, increased person–organization value congruence among employees is expected to be related to the ethical climate in Chinese SMEs. An organizational ethical climate affects employees’ perceptions and feelings about the organization; in general, an ethical climate reduces stress and imbalance in employees’ work. Thus, they are more willing to put efforts into the organization, adjust their values to the reality of the workplace, and increase the consistency of their values with the organization [49]. Moreover, when the ethical climate in an organization matches its employees’ personal ethical values, employees make efforts for the benefit of the organization. In addition, the moral climate in an organization affects the meaning of employees’ work and motivates them to perform better and produce value in their work [50]. In a strong ethical climate, employees feel the prevalence of ethical values and the clarity of norms and ethical standards within the organization and thus are strict with themselves in order to meet the organizational ethical standards, which creates a strong sense of organizational identification [10]. An organization with a strong ethical climate allows employees to feel ethical standards within the organization, improve their ethical values, and, in the long run, feel more clearly aligned with the values of the organization and promote organizational identification. Therefore, when the organization’s ethical climate is strong, the value consistency between employees and the organization is also improved, thus increasing their sense of identification with the organization. Moreover, when employees have a strong sense of identification with the organization, they have a positive attitude toward whistleblowing and believe that such behavior will be beneficial to the organization, which promotes whistleblowing [37]. An ethical climate with clear practices, standards, and procedures helps employees focus on their own ethical issues and encourages employees to clarify and maintain high ethical standards in the organization, increase organizational identification, and behave in a positive and ethical way for the development of the organization [51]. Therefore, this study highlights that employees’ person–organization value congruence moderates the mediating effect of organizational identification between organizational ethical climate and whistleblowing behavior. In an organization with a strong ethical climate, employees will have a deep understanding of ethical standards and combine the organization’s standards with their own. This creates a sense of attachment to the organization, stimulates the desire to protect the organization, and encourages resistance to unethical behavior through whistleblowing.

Therefore, this study uses organizational identification as a mediating variable between the organizational ethical climate and whistleblowing behavior, with person–organization value congruence as a moderating variable among ethical organizational climate, organizational identification, and whistleblowing behavior. This study emphasizes that person–organization value congruence moderates the mediating effect of employee organizational identification on Chinese SMEs. Based on the above theory, this study proposes the following hypotheses:

**Hypothesis** **2.**
*Person–organization value congruence will have a positive moderating effect on the relationship between organizational ethical climate and organizational identification. (The higher the person–organization value congruence value congruence, the stronger the influence of organizational ethical climate on organizational identification).*


**Hypothesis** **3.**
*The mediating influence of organizational identification on the relationship between organizational ethical climate and whistleblowing behavior will be moderated by person–organization value congruence. (The higher the level of person–organizational values congruence, the higher the impact of organizational ethical climate on whistleblowing behavior through organizational identity identification).*


### 2.3. The Moderated Mediation Effect of Leader Ethical Behavior

Leader ethical behavior refers to leadership behavior in which employees perceive their leaders as making ethical decisions [4]. Leader ethical behavior promotes the creation of ethical values, norms, and rules that make employees feel united in the organization [10]. Leader ethical behavior can also be expressed as the behavior of leaders who create moral values and principles [52]. Leadership roles affect employees and the organization; therefore, when leaders with ethical behavior set themselves as role models in the organization, they encourage and motivate employees to engage in similar behavior [30]. Research has found that ethical leadership is positively related to an ethical climate [24] and organizational citizenship behavior [4]. Based on these theories, this study argues that leader ethical behavior is morally meaningful behavior that leaders exhibit toward their employees and organizations with ethical awareness. Ethical leadership behavior can help employees better integrate into the organization and motivate them to realize their self-worth at work.

This study also emphasizes the moderating role of leaders’ ethical behavior and suggests that it can increase the role of the organization’s ethical climate in organizational identification and whistleblowing behavior. Therefore, the organizational identification and whistleblowing behavior of employees in Chinese SMEs are determined by the interaction between the organization’s ethical climate and leaders’ ethical behavior. Leaders with highly ethical behavior value and reward norms and policies of ethical behavior and ensure that organizational outcomes are influenced by an ethical work environment [10]. In addition, the leadership role affects both employees and the organization; therefore, when leaders set themselves as role models in the organization, they encourage and motivate employees to engage in certain ethical behaviors [30]. Therefore, ethical leadership behaviors lead to the convergence of employees’ self-identification and organizational identification, which, in turn, has a positive impact on employees’ organizational identification [53]. Positive and reliable leaders may play a key role in promoting whistleblowing behavior [54]. Leader ethical behavior is used as a benchmark to help shape employee behavior, encourage ethical decision making, and promote whistleblowing behavior by reducing the threat posed by retaliation [55]. Therefore, when ethical leadership behavior is strong, employees’ organizational identification and whistleblowing behaviors also increase. This suggests that ethical leadership behavior has a positive impact on Chinese SMEs. Under this influence, organizational identification and whistleblowing behavior among employees increases. Therefore, when the standard of ethical leadership behavior is high in Chinese SMEs, employees use the leader as a role model and increase their own concern for organizational ethics. This increases organizational identification and whistleblowing behavior.

Additionally, leaders’ ethical behavior enhances organizational and employee behavioral management, thus contributing to the formation of an ethical climate [56]. Therefore, the level of ethical leadership behavior among Chinese SME employees is likely related to the organizational ethical climate. Leaders’ ethical behavior contributes to the formation of an ethical climate by demonstrating ethical decision making, communicating ethically with employees regularly, and clarifying to employees that maintaining ethics is an important organizational outcome [57]. If leaders act in an honest and trustworthy manner, these behaviors create a virtuous cycle in which ethical leadership behaviors perpetuate an ethical work environment that allows for prosperity [58]. Therefore, this study highlights that leaders’ ethical behavior in SMEs improves the organization’s ethical climate. In addition, the ethical climate of an organization indicates that the organization cares about everyone and when employees feel that the organization cares, they are obliged to reciprocate in some way, which increases their trust in the organization and identification with it [59]. Moreover, in organizations with a strong ethical climate, leaders shape individual employees’ ethical values and beliefs through continuous ethical education, which leads to whistleblowing behavior and eliminates and prevents unethical behaviors [60]. Therefore, an ethical organizational climate’s role in improving organizational identification and whistleblowing behavior among employees is more certain in SMEs. Ethical leadership behavior also plays an important role in transmitting ethical values to employees, fostering an ethical climate within the company, creating a bond with the organization, and promoting organizational identification [24]. When the leader is characterized by ethicality, employees use the leader as a role model, are more willing to follow suit, and have a positive attitude toward whistleblowing behavior for the benefit of the organization [61]. Therefore, when the standard of ethical leadership behavior in an organization is high, employees’ organizational identification and whistleblowing behavior also remain high. If leaders try to create responsible helpful rule-based types of organizational climate, they also help employees integrate well into the organization [10]. Therefore, the higher the leaders’ ethical behavior in SMEs, the greater the effect of the organization’s ethical climate on employees’ organizational identification and whistleblowing behavior. This study highlights that ethical behavior moderates the mediating effect of organizational identification between the organizational ethical climate and whistleblowing behavior. Therefore, in an organizational ethical climate, leaders make ethical performance more prominent, promote employees’ attention to and understanding of ethical norms, strengthen self-ethical awareness, increase employees’ organizational identification, and contribute to employees’ willingness to engage in whistleblowing to reduce unethical behavior in the organization.

This study uses organizational identification as a mediating variable between ethical organizational climate and whistleblowing behavior. Ethical leadership behavior serves as another moderating variable between organizational ethical climate, organizational identification, and whistleblowing behavior. This study emphasizes that ethical leadership behavior moderates the mediating effect of employees’ organizational identification in Chinese SMEs. Based on the above theories, this study proposes the following hypotheses:

**Hypothesis** **4.**
*Leader ethical behavior will have a positive moderating effect on the relationship between organizational ethical climate and organizational identification. (The higher the level of leader ethical behavior, the higher the impact of organizational ethical climate on organizational identification).*


**Hypothesis** **5.**
*Leader ethical behavior will have a positive moderating effect on the relationship between organizational ethical climate and whistleblowing behavior. (The higher the level of leader ethical behavior, the higher the impact of organizational ethical climate on whistleblowing behavior).*


**Hypothesis** **6.**
*The mediating influence of organizational identification on the relationship between organizational ethical climate and whistleblowing behavior will be moderated by leader ethical behavior. (The higher the level of leader ethical behavior, the higher the impact of organizational ethical climate on whistleblowing behavior through organizational identity identification).*


Table 1 presented below summarizes the hypotheses of this paper.

## 3. Methods

### 3.1. Sample Characteristics

This study examined the behaviors of organization members working in Chinese SMEs and to gather data, an online questionnaire survey was conducted. A total of 344 responses were collected and used for the empirical analysis. Regarding the demographic characteristics of this study, 139 (40.4%) males and 205 (59.6%) females participated. Regarding age, 1 (0.3%) person was under 20 years old, 57 (16.6%) were 20 to 29 years old, 76 (22.1%) were 30 to 39 years old, 115 (33.4%) were 40 to 49 years old, and 95 (27.6%) were 50 years or older. In the realm of education, among the surveyed individuals, 114 (33.1%) had attained technical secondary or high school qualifications, 73 (21.2%) held junior college diplomas, 78 (22.7%) had completed their college studies, 16 (4.7%) had achieved master’s degrees, 3 (0.9%) had obtained doctoral degrees, and 60 (17.4%) had pursued alternative educational paths. Concerning employment status, the majority, comprising 211 (61.3%) respondents, were engaged in full-time positions, while 133 (38.7%) were employed in informal capacities.

Regarding service years, 27 (7.8%) had been working for a year or less, 44 (12.8%) had been working for 1 to less than 3 years, 45 (13.1%) had been working for 3 to less than 5 years, 38 (11.0%) had been working for 5 to less than 7 years, and 190 (55.2%) had been working for 7 or over.

In terms of tenure with their current immediate leader, the breakdown is as follows: 48 (14.0%) individuals had worked together for a year or less, 41 (11.9%) for 1 to 2 years or less, 55 (16.0%) for 2 to 3 years or less, 32 (9.3%) for 3 to 4 years or less, 31 (9.0%) for 4 to 5 years or less, and 137 (39.8%) for 5 or more years.

Regarding enterprise type, 30 (8.7%) of the individuals worked in the education sector, 37 (10.8%) were employed in finance, 20 (5.8%) were associated with the medical industry, catering services employed 78 (22.7%) of the individuals, coal mining accounted for 39 (11.3%) of the workforce, 8 (2.3%) were engaged in media-related activities, and the remaining 132 (38.4%) were occupied in various other fields. 

Table 2 presented below is the analysis result table of the demographic characteristics.

### 3.2. Measurement

Organizational ethical climate refers to the shared views between employees and the organization about ethical behavior and how to solve ethical problems [62]. To measure Chinese SMEs’ organizational ethical climate, this study used a tool mentioned in the studies of [63]. The measurement tool consists of nine items. Sample items included “People in this company strictly obey the company policies”, “It is very important to follow the company’s rules and procedures here”, “People are expected to do anything to further the company’s interests”, and “People are concerned with the company’s interests to the exclusion of all else”.

Organizational identification is the extent to which employees identify with and feel a sense of belonging to their organization [64]. To measure Chinese SMEs’ organizational identification, this study used a tool mentioned in the studies [65]. The measurement tool consisted of six items that participants responded to sample items included statements such as “‘When someone praises (name of school), it feels like a personal compliment” and “When someone criticizes (name of school), it feels like a personal insult”. “I am very interested in what others think about (name of school)” and “This school’s successes are my successes”.

Person–organization value congruence refers to employees who work in an organization who not only align with the organization’s values but also uphold and cherish those values [66]. To measure Chinese SMEs person–organization value congruence, this study used a tool mentioned in the studies [66]. The measurement tool consists of five items. Sample items included “I agree with the values of my organization”, “My per-sonal values match the values of my organization”, “My organization’s values and culture provide a good fit with the things that I value in life”, and “I find that sometimes I have to compromise personal principles to conform to my organization’s expectations”.

Leader ethical behavior refers to the leader’s own values, attitudes, decision making, and influence processes and how these processes affect employee behavior [67]. To measure Chinese SME leaders’ ethical behavior, this study used a tool mentioned in the studies of [68]. The measurement tool consists of eight items. Sample items included “Had the best interests of an employee in mind”, “Made a fair and balanced decision”, “Listened to what an employee had to say”, and “Disciplined an employee who violated ethical standards”.

Whistleblowing occurs when an organization member (former or current) discloses illegal, unethical, or illegal conduct under the control of his or her employer to a person or organization who may take action [69]. To measure Chinese SMEs’ whistleblowing behavior, this study used a tool mentioned in the studies of [70]. The measurement tool consists of seven items. Sample items included “I would report it through channels outside of the organization”, “I would report it to my immediate supervisor”, “I would disclose it by going public”, and “I would report it by using internal procedures”.

We used a 7-point Likert scale (ranging from 1 = strongly disagree to 7 = strongly agree). Figure 1 presented below is the research model of this paper.

## 4. Results

### 4.1. Confirmatory Factor Analysis and Reliability Analysis

In this study, the feasibility of various data models is assessed using confirmatory factor analysis [71]. The outcomes derived from confirmatory factor analysis are presented below. The absolute fit indices yielded the following results: *X*²(*p*) = 2158.801(0.000), *X*²/df = 4.050, and RMSEA = 0.094. The RMSEA is considered a measure of the overall fit quality, where values closer to 0 indicate a better fit and larger values suggest a poorer fit. Regarding the RMSEA, values lower than 0.05 indicate a minimal approximation error, values ranging from 0.05 to 0.08 suggest an acceptable level of approximation error, and values higher than 0.10 indicate a substantial discrepancy between the model and the observed data, indicating poor fit [72]. The incremental fit indices, including IFI = 0.924 and CFI = 0.923, indicate the extent to which the proposed model improves upon a null model. On the other hand, the parsimonious adjusted indices, namely PNFI = 0.807 and PGFI = 0.616, assess the goodness of fit while considering the complexity of the model.

In this study, the average variance extracted (AVE) and composite reliability (C.R) values were analyzed. The AVE values for different constructs were as follows: organizational ethical climate (0.801), organizational identification (0.833), person–organization value congruence (0.865), leader ethical behavior (0.833), and whistleblowing behavior (0.756). All these values exceeded the threshold of 0.5, indicating satisfactory convergent validity.

When considering the composite reliability (C.R), the values obtained for each construct were as follows: organizational ethical climate (0.967), organizational identification (0.958), person–organization value congruence (0.961), leader ethical behavior (0.967), and whistleblowing behavior (0.916). All of these values exceeded the threshold of 0.7, indicating strong internal consistency and reliability.

Reliability analysis refers to the method of measuring the internal consistency of scale items [73]. Therefore, in this study, Cronbach’s alpha coefficient was employed for this purpose. The calculated values for Cronbach’s alpha were as follows: organizational ethical climate (0.973), organizational identification (0.970), person–organization value congruence (0.973), leader ethical behavior (0.979), and whistleblowing behavior (0.978). To establish the reliability of the measurement, it is commonly accepted that Cronbach’s alpha should be higher than 0.7. In this study, all variables met this criterion, indicating strong internal consistency. The results are shown in Table 3.

### 4.2. Descriptive Statistics and Correlation Analysis

Table 4 shows the descriptive statistics and correlation analysis. Descriptive statistics analysis included the mean and standard deviation (SD). The means for organizational ethical climate, organizational identification, person–organization value congruence, leader ethical behavior, and whistleblowing behavior were 5.908, 5.729, 5.686, 5.738, and 4.880, respectively. In addition, the SDs of organizational ethical climate, organizational identification, person–organization value congruence, leader ethical behavior, and whistleblowing behavior were 0.986, 1.126, 1.145, 1.137, and 1.733, respectively.

To verify the correlation among variables, this study conducted a correlation analysis; the results are summarized as follows: organizational ethical climate was positively associated with organizational identification (*r* = 0.897, *p* < 0.001), person–organization value congruence (*r* = 0.862, *p* < 0.001), leader ethical behavior (r = 0.909, *p* < 0.001), and whistleblowing behavior (*r* = 0.477, *p* < 0.001). Organizational identification was positively associated with person–organization value congruence (*r* = 0.917, *p* < 0.001), leader ethical behavior (r = 0.899, *p* < 0.001), and whistleblowing behavior (r = 0.569, *p* < 0.001). Person–organization value congruence was positively associated with leader ethical behavior (*r* = 0.918, *p* < 0.001) and whistleblowing behavior (*r* = 0.573, *p* < 0.001). Moreover, leader ethical behavior was positively associated with whistleblowing behavior (*r* = 0.548, *p* < 0.001).

### 4.3. Path Analysis

SPSS Process Model 4 was used to analyze the mediation effect of organizational identification. The results show that organizational ethical climate has a positive impact on organizational identification (Estimate = 1.023, *p* < 0.001). But organizational ethical climate was not shown to have a positive impact on whistleblowing behavior (Estimate = −0.300, *p* > 0.05). In addition, the results show that an organizational identification has a positive impact on whistleblowing behavior (Estimate = 1.112, *p* < 0.001). Hypothesis 1 found that the influence of organizational ethical climate on whistleblowing behavior is mediated by organizational identification. The indirect effect was found to be 1.139. The bootstrapped confidence intervals were calculated to be Boot LLCI = 0.832 and Boot ULCI = 1.502. Since the interval does not include the value of zero, it suggests a statistically significant indirect effect. These findings suggest that there was a significant mediation effect of organizational identification. These results indicate that organizational ethical climate had a positively impact on whistleblowing behavior, and this effect was mediated by organizational identification. Therefore, Hypothesis 1 is supported. Detailed information regarding the path analysis results can be found in Table 5.

### 4.4. Moderating Effect of Person–Organization Value Congruence

Hypothesis 2 demonstrated that the moderating effect of person–organization value congruence was observed in relation to the influence of organizational ethical climate on organizational identification. The findings indicated a significant moderation effect (*β* = 0.047, *p* < 0.05), suggesting that the impact of organizational ethical climate on organizational identification is amplified when there is higher person–organization value congruence. This supports hypothesis 2 and suggests that the interaction between person–organization value congruence and organizational ethical climate contributes to a stronger sense of organizational identification. In essence, these results imply that a greater alignment between individual and organizational values enhances the relationship between organizational ethical climate and organizational identification. Table 6 presented below is the moderating effect of person–organization value congruence on the relationship between organizational ethical climate and organizational identification.

Figure 2 presented below is pragh of moderating effect of person–organization value congruence on the relationship between organizational ethical climate and organizational identification.

### 4.5. Moderated Mediation Effect of Person–Organization Value Congruence

Table 7 shows the moderated mediation effect on person–organization value congruence. Hypothesis 3 suggests that person–organization value congruence moderated the mediating influence of organizational identification on the relationship between organizational ethical climate and whistleblowing behavior. To test this hypothesis, we were examined using SPSS PROCESS Macro 3.4.1 Model 7 and was tested using 95% confidence intervals and 5000 bootstrapping re-samples.

To conditional indirect effect of organizational ethical climate on whistleblowing behavior, the index of the moderated relationship at three different levels of the moderator variable: −1 SD, mean (M), and +1 SD. Since 0 was not included be-tween Boot LLCI and Boot ULCI at the level of −1 SD (standard deviation), mean level (M), and mean +1 SD (standard deviation) confidence intervals, it was concluded that statistical significance was confirmed.

Furthermore, the index of moderated mediation values was 0.0263, Boot SE = 0.0155, Boot LLCI = 0.0033, and Boot ULCI = 0.0644. As 0 was not included between Boot LLCI and Boot ULCI, this proved that the bootstrapped confidence interval was significant. Therefore hypothesis 3 was supported.

### 4.6. Moderating Effect of Leader Ethical Behavior

Hypothesis 4 established that leader ethical behavior moderated the effect of organizational ethical climate on organizational identification. The findings indicate that leader ethical behavior significantly moderated the effect of organizational ethical climate on organizational identification (*β* = 0.054, *p* < 0.05). This means that the higher the leader ethical behavior, the greater the impact of organizational ethical climate on organizational identification. Therefore, hypothesis 4 is supported and through the results, we can find that the interaction between leader ethical behavior and the organizational ethical climate will lead to a higher degree of organizational identification. Table 8 presented below is pragh moderating effect of leader ethical behavior on the relationship between organizational ethical climate and organizational identification.

Figure 3 presented below is pragh of moderating effect of leader ethical behavior on the relationship between organizational ethical climate and organizational identification.

Hypothesis 5 suggests that the influence of organizational ethical climate on whistleblowing behavior is moderated by leader ethical behavior. The findings indicate that there was a significant moderation effect of leader ethical behavior on the relationship between organizational ethical climate and whistleblowing behavior (*β* = 0.105, *p* < 0.1). In simpler terms, when leaders demonstrate higher ethical behavior, the impact of the organizational ethical climate on whistleblowing behavior becomes stronger. Thus, the results support Hypothesis 5, indicating that the interaction between leader ethical behavior and the organizational ethical climate leads to an increased likelihood of whistleblowing behavior. Table 9 presented below is pragh moderating effect of leader ethical behavior on the relationship between organizational ethical climate and whistleblowing behavior.

Figure 4 presented below is pragh moderating effect of leader ethical behavior on the relationship between organizational ethical climate and whistleblowing behavior.

### 4.7. Moderated Mediation Effect of Leader Ethical Behavior

Table 10 shows the moderated mediation effect on leader ethical behavior. Hypothesis 6 established that leader ethical behavior moderated the mediating influence of organizational identification on the relationship between organizational ethical climate and whistleblowing behavior. The moderated mediation model was examined using SPSS PROCESS Macro 3.4.1 Model 7 and was tested using 95% confidence intervals and 5000 bootstrapping re-samples.

To assess the conditional indirect effect of organizational ethical climate on whistleblowing behavior, the index of the moderated relationship at three different levels of the moderator variable: −1 SD, mean (M), and +1 SD. Since 0 was not included be-tween Boot LLCI and Boot ULCI at the level of −1 SD (standard deviation), mean level (M), and mean +1 SD (standard deviation) confidence intervals, it was concluded that statistical significance was confirmed.

Furthermore, the index of moderated mediation values was 0.0289, Boot SE = 0.0162, Boot LLCI = 0.0066, and Boot ULCI = 0.0718. As 0 was not included between Boot LLCI and Boot ULCI, this proved that the bootstrapped confidence interval was significant. Therefore hypothesis 6 was supported.

## 5. Discussion

This study considers employees of Chinese SMEs as the research object and verifies the impact of an organizational ethical climate on whistleblowing behavior. A positive organizational climate will make employees feel that the organization cares about them and strengthen their perception of positive emotions, which will resonate with the organization [17]. Specifically, this study suggests that an organizational ethical climate that expresses fairness, protects employees’ interests, and emphasizes ethical norms will make employees feel respected and valued in the organization, which will make employees take the organization’s goals as their own and gradually create a sense of identification with the organization. When employees agree with the organization’s values, they will have a sense of organizational identity and be willing to contribute to the organization [53]. Therefore, when the interests of the organization are harmed, employees will take the initiative to stand up to protect the interests of the organization. Related to this, the organizational ethical climate will stimulate employees’ sense of organizational identity, which will lead them to engage in whistleblowing behaviors. By detailing the mediating role of organizational identification, this study expands the field of research on organizational ethical climate by providing a new perspective. In addition, this study used person–organizational value congruence and leader ethical behavior as moderating variables and verified their moderating effects. The findings suggest that organizational ethical climate increases employees’ organizational identification in the interaction of individual–organizational values congruence and leader ethical behaviors and that leader ethical behaviors moderated the effects between organizational ethical climate and whistleblowing behaviors. Meanwhile, we validated the moderated mediation model to determine the role of ethical climate in organizations and its positive impact on organizations to improve their sustainability and viability. Finally, the results of this study provide additional theoretical perspectives for future research.

The empirical analysis results provide inspiration for future research and sustainable development of Chinese SMEs and indicate development directions. The conclusions are summarized below.

### 5.1. Theoretical Implications

The main objective of this study is to explore and determine how an ethical organizational climate leads to whistleblowing. It focuses on the impact of an ethical organizational climate on whistleblowing behavior but also specifically studies which key variables play a role in an organization’s ethical climate process by inducing employees’ whistleblowing behavior.

First, a strong ethical climate within an organization enhances the level of organizational identification among employees. This means that when an organization’s ethical climate is robust, employees feel a stronger sense of connection and affiliation with the organization. The ethical climate of an organization serves as a mechanism for promoting alignment between employees’ attitudes and behaviors and those that are advantageous for the organization [50]. When the ethical climate is strong, it enhances employees’ awareness of ethical considerations and fosters a commitment to behaving in morally upright ways both inside and outside of work [8]. Moreover, when employees have a high degree of identification with the organization, the organization’s values, norms, and interests are better integrated into its self-concept [41]. Therefore, when employees feel a strong moral atmosphere within the organization, they will be clear about the organization’s ethical standards, have increased trust in the organization, and promote organizational identification among employees. Therefore, among Chinese SMEs, the stronger the ethical climate in the organization, the higher the employees’ identification with it.

Second, in this study, the organization’s ethical climate had no impact on employee whistleblowing behavior. This shows that the level of an organization’s ethical climate does not affect whistleblowing behavior. Whistleblowing behavior benefits an organization’s ethics and risks [8]. However, employees curb unethical behavior through whistleblowing for the benefit and long-term development of the organization. Employees in organizations tend to be more hesitant about whistleblowing behaviors, mainly fearing that they may incur adverse effects and consequences [61]. Specifically, whistleblowing behavior makes other employees perceive that their behavior is being monitored and their own activities are threatened, which can easily stimulate negative cognitive evaluations of other employees and trigger negative emotional reactions [1]. Also, employees with a high level of individualism may perceive their self-interests to be higher than the interests of the organization, even though they work in a unit with an ethical atmosphere, making them more reluctant to show whistleblowing behaviors [74]. Therefore, when employees in an organization feel that their personal interests take precedence over the interests of the organization, even if they work in a workplace with an ethical climate, they will not resonate with the organization’s sense of morality and they will choose not to report their behavior out of self-interest and self-protection considerations.

Third, organizational identification has a positive impact on whistleblowing behavior. This shows that the stronger employee identification with the organization, the more employee whistleblowing behavior. Organizational identification is an important construct that explains behaviors related to organizational outcomes [10]. Employees with strong organizational identification behave beyond their own interests for the overall benefit of the organization [27]. In addition, if the unethical behavior of the organization infringes upon the public interest, then, from an ethical point of view, whistleblowing behavior can be regarded as altruistic and socially desirable, so reporting the behavior aims to correct negative outcomes and benefit the organization and society [75]. Therefore, an employee with strong identification with the organization quickly integrates into organizational life, strives to achieve organizational goals, and holds a positive attitude toward reporting behavior for the sustainable development of the organization.

Fourth, organizational identification has a complete mediating effect between ethical organizational climate and whistleblowing behavior. This shows that an ethical organizational climate can only impact whistleblowing behavior through organizational identification. In an organization with an ethical climate, a positive working environment can make employees feel warm and realize their self-worth within the environment, thereby increasing organizational identification, spontaneously stimulating protection of the organization and expanding organizational identification-driven results, leading to whistleblowing [76]. An ethical climate can enhance employees’ awareness of ethical obligations, which not only prevents unethical behavior but also enhances their ability to talk about organizational issues [77]. In Chinese SMEs, employees are likely to experience high-intensity work activities and, although the enterprises have been developed rapidly, employees face great stress and emotional resource depletion due to the imbalance between payment and income [78,79]. In such an environment, a positive organizational climate will make employees feel that the organization cares and reduce their stress and emotional consumption [17]. On the contrary, employees with a higher sense of organizational identity will show a strong sense of belonging to the organization, identify with their organization in attitude, have a strong sense of responsibility, and then produce behaviors that are beneficial to the organization [80]. Previous research has shown that when employees have strong organizational identification, they will have positive attitudes and evaluations of the organization, which in turn affects their personal attitudes and behaviors toward their work [81]. Therefore, in this climate, employees have increased trust in the organization, an improved sense of attachment to the organization, and strengthened identification with the organization. They then actively prevent unethical behavior in the organization through whistleblowing. This attempt helps to expand the field of research and reveals how organizational ethical climate has an impact on whistleblowing behavior through organizational identity. This provides a basis for future research.

Fifth, this study verifies the moderating effect of employee person–organization value congruence on the relationship between ethical climate and organizational identification. The results show that person–organization value congruence has a positive moderating effect on the relationship between ethical organizational climate and organizational identification. This shows that the higher the interaction between employees’ person–organization value congruence and the organizational ethical climate, the higher the organizational identification. The organization’s ethical climate affects employees’ identification with the organization [82]. Moreover, consistency between personal and organizational spiritual values makes employees more firmly identify with their organization, leading them to be more committed to their work [46]. Therefore, employees who work in an ethical organizational climate are more likely to have their own beliefs and goals and work hard to complete the organization’s objectives. Being affected by the organizational climate for a long time will enhance their perceptions of the consistency between organizational values and increase their organizational identification.

Sixth, this study verified the moderating effect of leaders’ ethical behavior on the relationship between organizational ethical climate, organizational identification, and whistleblowing behavior. The results show that ethical leadership behavior has a positive moderating effect on the relationships between organizational ethical climate, organizational identification, and whistleblowing behavior. This shows that the higher the interaction between leaders’ ethical behavior and the organization’s ethical climate, the higher the level of organizational identification and whistleblowing behavior. Leaders’ moral level is an important factor affecting the formation of an ethical climate. When employees perceive a leader to have ethical behavior, the psychological distance between the employee and the organization can be shortened, employees’ sense of belonging and identification with the organization can be enhanced, and employees’ emotional commitment to provide positive feedback to the organization can be promoted [82]. Moreover, ethical leadership in an organization affects employees’ moral identity and encourages them to expose unethical behaviors in the work environment [30]. Therefore, leaders’ ethical behavior can help organizations shape an ethical climate by encouraging employees to have better work experiences, increase their sense of attachment to the organization, increase organizational identification, abide by organizational ethics, reduce the occurrence of self-immoral behavior, serve the interests of the organization, and prevent things that harm the interests of the organization through reporting.

Finally, we aimed to prove whether person–organization value congruence and leaders’ ethical behavior moderated the mediation effect of organizational identification. This study verified the moderated mediating role of person–organization value congruence and ethical leadership behavior. The results show that both person–organization value congruence and ethical leadership behavior have significant moderating mediation effects. This means whistleblowing behavior is moderated by the interaction between the organization’s ethical climate and person–organization value congruence, as well as the interaction between the organization’s ethical climate and leaders’ ethical behavior. To verify the moderated mediation effect of person–organization value congruence and ethical leadership behavior, a more comprehensive method was used to induce whistleblowing behavior. This provides a basis for future exploration and identification of more effective methods for inducing whistleblowing behavior.

### 5.2. Practical Implication

First, an organizational ethical climate is a common experience and perception of ethical issues within an organization [7]. A strong ethical climate conveys positive ethical organizational values and behavioral standards [8]. Moreover, employees in a positive working climate are more willing to propose solutions to problems, improvement plans, constructive ideas, the need for innovation, and the pursuit of innovation [83]. Therefore, in management practice, organizations should create an ethical climate, strengthen employees’ awareness of moral obligations, increase ethical behavior, and constantly motivate employees at work, prompting them to voluntarily contribute to the organization and have a positive impact on organizational behavior.

Second, the organizational ethical climate management model can enhance employees’ professional identity and solidarity, as well as improve their psychological health and job satisfaction [16]. Therefore, while actively creating an ethical climate in the organization, the organization should strengthen the management of employees, resisting individualistic behaviors in the organization, increasing the implementation of rules and regulations by employees, and promoting employees to give back to the organization and help others in an ethical manner.

Third, leaders with moral identification, ethical values, and ethical awareness can help the organization better establish an ethical climate [10]. The characteristics of ethical leadership allow employees to perceive fairness and affirmation in the organization and transform this perception into identification with the organization, emotional attachment, and belonging [82]. Therefore, organizations should focus on cultivating ethical behavioral norms among leaders, improving their effectiveness, and encouraging them to exhibit more ethical leadership behaviors, thereby promoting employees’ positive attitudes toward the organization.

Fourth, organizational identification can be defined as a sense of unity that can be enhanced through communication within the workplace [46]. Moreover, organizational identification is also a strong psychological connection between an individual and the organization [84]. Therefore, in management practices, organizations should pay attention to and strengthen communication and connections between leaders and employees. Leaders should create a positive working climate, encourage employees to participate in the organization’s collective activities, encourage employees to quickly integrate into organizational life, and improve employees’ identification with the organization.

Fifth, an organization’s ethical culture and a leader’s personal ethics resonate. They have highly similar value bases and leaders play an important role in shaping the ethical culture of the company, the ethical climate of the organization, and influencing the behavior of employees [85]. Moreover, as the core elements of an organization, leaders’ behaviors and values greatly influence the behaviors and attitudes of employees. Therefore, organizations should focus on developing the moral and ethical qualities and behaviors of leaders, strengthening leaders to establish a clear code of ethics and personal values in the organization and improving leaders’ personal ethical behavior in the organization. At the same time, leaders should be encouraged to learn more about ethics outside of work and the organization can provide training to develop ethical behaviors and qualities in leaders.

Finally, many factors influence employees’ whistleblowing behavior. In addition to personal characteristics, other aspects contribute, such as support from leaders or colleagues, corporate culture, and the environment [33]. In addition, whistleblowing affects employees’ future behavior. Although reporting can correct unethical behavior in the organization and enhance self-worth, in the long run, whistleblowing behavior will turn every employee into a potential monitor of others and create close supervision. This leads to a sense of distrust and alienation among employees [37]. Therefore, leaders should strengthen employees’ ethical behavioral norms, establish positive ethical values, and encourage employees to report violations in the organization. At the same time, more attention should be paid to the psychological state of bystanders when whistleblowing occurs, reducing their psychological pressure and increasing their moral identity for whistleblowing behavior, thereby preventing retaliation among peers.

### 5.3. Limitations and Future Research

Although this study helps verify the impact of organizational identification on the relationship between organizational ethical climate and whistleblowing behavior and as a moderated mediator of person–organization value congruence and leader ethical behavior, it is also limited to a certain extent, as detailed below.

First, this study examines the role of organizational ethical climate and whistleblowing behavior in Chinese SMEs. According to prior research, organizational ethical climate was found to have a positive effect on whistleblowing behavior [86,87]. Whereas, there is no direct effect in this study. Due to different cultural differences in different countries, geographical location and culture should be taken into account. Therefore, in future studies, it is necessary to conduct similar studies on members of organizations from different countries and cultures. This is because it is of great research significance to see whether the same results can be derived when empirically analyzing organizational members from different countries and cultures [88]. It can also be compared by integrating the results of the studies.

Second, this study regards whistleblowing as spontaneous positive employee behavior. However, based on pressure transaction theory, it is also regarded as a source of stress and can both aggravate and inhibit uncivilized behavior in the workplace. Specifically, whistleblowing aggravates workplace incivility by affecting the attributions of hostility [1]. Therefore, future research should focus on reporting as a negative behavior to explore its impact on an organization.

Third, in this study, an organization’s ethical climate had no effect on whistleblowing behavior. This shows that the organization’s ethical climate does not directly affect whistleblowing behavior. Future research should explore the impact of other independent variables on whistleblowing behavior.

Fourth, the relatively small sample size of this study has certain limitations that may lead to biased results and the corresponding sample size will be subsequently increased to improve the accuracy of the data. Moreover, this study was only conducted on SMEs, which may also have an impact on other industries and the nature of other units. Therefore, in future research, it is necessary to consider the research object in the medical, nursing, police, IT, and other industries for the study.

Fifth, PLS (Partial Least Squares) is a technique that is currently being actively used and shows stronger effectiveness for verifying mediation effects. In order to further secure the effectiveness of the mediation effect in this study, it was necessary to use the PLS technique. However, the failure to use PLS is a limitation of this study. A recent previous study also used the PLS technique [89,90,91]. In future research, the mediation effect should be verified using the PLS technique rather than the existing universal mediation effect verification technique.

Sixth, this study did not establish classification and most participants in the survey were employees. We believe this approach leads to overly correlated variables. Therefore, we believe future research should incorporate a common method bias (CMB). Management of questionnaires should be strengthened and surveys should be conducted among employees on leadership issues and leaders should address employees’ attitudes, behaviors, performance, and other issues [88], thereby increasing the value of data. 

Finally, we designed and conducted a cross-sectional study to analyze survey data from China. However, a cross-sectional study may not rule out various potential reverse impacts or roles. Such a problem can still be seen as a limitation. In order to derive more specific and clear causal relationships and results, a longitudinal study will need to be conducted in the future.

## Figures and Tables

**Figure 1 behavsci-14-00293-f001:**
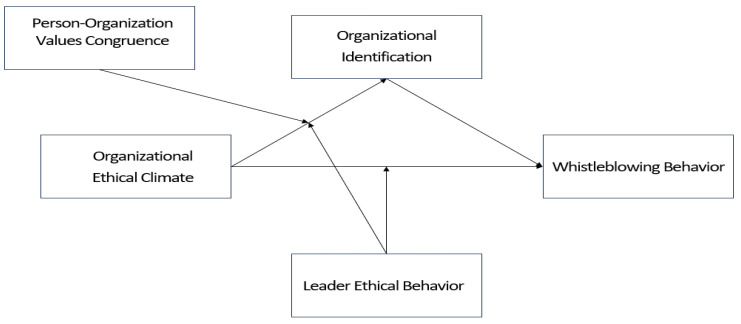
Research model.

**Figure 2 behavsci-14-00293-f002:**
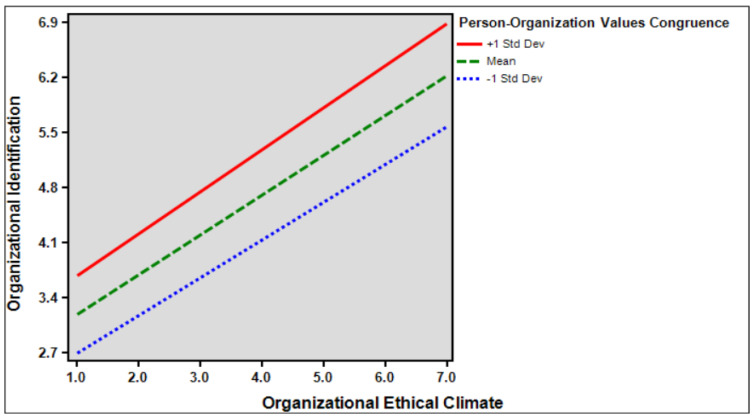
The moderating effect of person–organization value congruence.

**Figure 3 behavsci-14-00293-f003:**
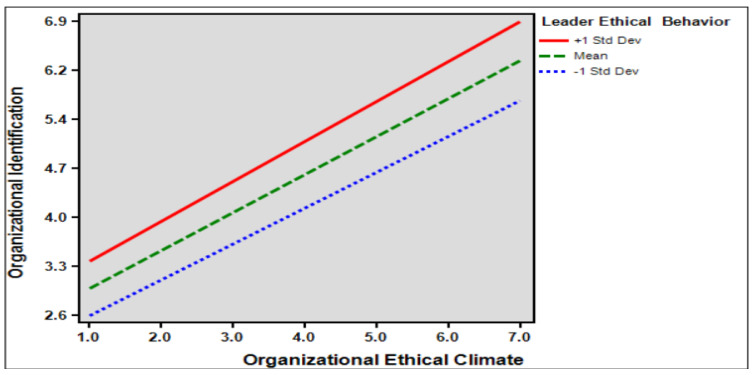
The moderating effect of Leader Ethical Behavior.

**Figure 4 behavsci-14-00293-f004:**
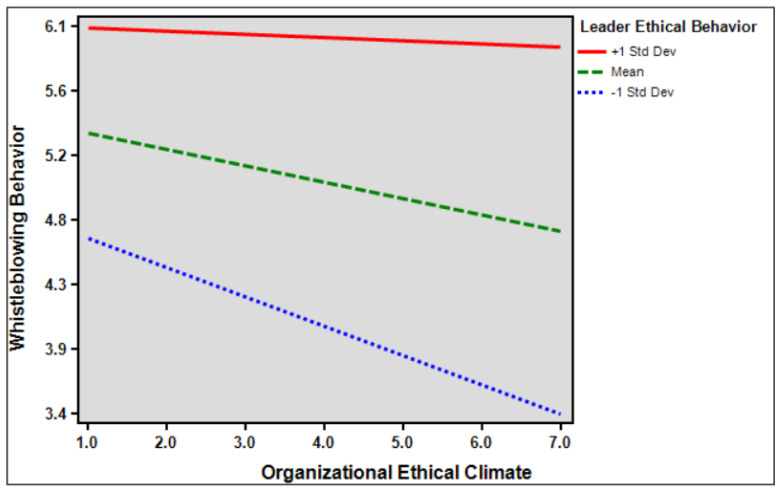
The moderating effect of leader ethical behavior.

**Table 1 behavsci-14-00293-t001:** Hypotheses.

H1	Organizational identification will mediate the relationship between organizational ethical climate and whistleblowing behavior.
H2	Person–organization value congruence will have a positive moderating effect on the relationship between organizational ethical climate and organizational identification. (The higher the person–organization value congruence value congruence, the stronger the influence of organizational ethical climate on organizational identification).
H3	The mediating influence of organizational identification on the relationship between organizational ethical climate and whistleblowing behavior will be moderated by person–organization value congruence. (The higher the level of person–organizational values congruence, the higher the impact of organizational ethical climate on whistleblowing behavior through organizational identity identification).
H4	Leader ethical behavior will have a positive moderating effect on the relationship between organizational ethical climate and organizational identification. (The higher the level of leader ethical behavior, the higher the impact of organizational ethical climate on organizational identification).
H5	Leader ethical behavior will have a positive moderating effect on the relationship between organizational ethical climate and whistleblowing behavior. (The higher the level of leader ethical behavior, the higher the impact of organizational ethical climate on whistleblowing behavior).
H6	The mediating influence of organizational identification on the relationship between organizational ethical climate and whistleblowing behavior will be moderated by leader ethical behavior. (The higher the level of leader ethical behavior, the higher the impact of organizational ethical climate on whistleblowing behavior through organizational identity identification).

**Table 2 behavsci-14-00293-t002:** Sample characteristics.

		Quantities	Percentage (%)
Genders	Males	139	40.4%
	Females	205	59.6%
Age	Under 20 years old	1	0.3%
	20–29	57	16.6%
	30–39	76	22.1%
	40–49	115	33.4%
	50 or over	95	27.6%
Education	Technical secondary school or high school	114	33.1%
	Junior college	73	21.2%
	college graduates	78	22.7%
	Master’s degrees	16	4.7%
	Doctor’s degrees or over	3	0.9%
	other education	60	17.4%
Employment relationships	full-time jobs	211	61.3%
	informal positions	133	38.7%
Service Year	Worked for a year or less	27	7.8%
	1 to less than 3 years	44	12.8%
	3 to less than 5 years	45	13.1%
	5 to less than 7 years	38	11.0%
	More than 7 years or over	190	55.2%
Working with the current immediate leader	a year or less	48	14.0%
	1 to 2 years or less	41	11.9%
	2 to 3 years or less	55	16.0%
	3 to 4 years or less	32	9.3%
	4 to 5 years or less	31	9.0%
	5 or more years	137	39.8%
	worked in the education sector	30	8.7%
	employed in finance	37	10.8%
	associated with the medical industry	20	5.8%
Enterprise type	catering services	78	22.7%
	coal mining accounted	39	11.3%
	engaged in media-related activities	8	2.3%
	occupied in various other fields.	132	38.4

**Table 3 behavsci-14-00293-t003:** The results of confirmatory factor analysis and reliability analysis.

Variables		Estimate	S.E.	C.R.	*p*	Standardized Regression Weights	AVE	C.R	Cronbach’s Alpha
Organizational Ethical Climate (A)	A9	1				0.902	0.801	0.967	0.973
	A8	0.989	0.025	39.506	***	0.95			
	A7	0.913	0.025	36.542	***	0.935			
	A6	0.946	0.025	37.635	***	0.941			
	A5	1.027	0.041	25.2	***	0.836			
	A4	1.055	0.04	26.056	***	0.846			
	A3	0.901	0.031	28.8	***	0.877			
	A2	1.018	0.029	35.245	***	0.926			
	A1	0.816	0.033	25.066	***	0.835			
Organizational Identification (B)	B6	1				0.892	0.833	0.958	0.970
	B5	1.113	0.028	39.207	***	0.935			
	B4	0.951	0.03	31.273	***	0.884			
	B3	1.153	0.026	45.186	***	0.961			
	B2	1.024	0.031	32.751	***	0.895			
	B1	1.05	0.03	34.531	***	0.908			
Person–Organization Value Congruence (C)	C5	1				0.885	0.865	0.961	0.973
	C4	1.087	0.027	40.831	***	0.943			
	C3	1.081	0.026	42.35	***	0.95			
	C2	1.034	0.026	40.392	***	0.941			
	C1	1.075	0.03	35.968	***	0.931			
Leader Ethical Behavior (D)	D8	1				0.92	0.833	0.967	0.979
	D7	1.043	0.024	43.617	***	0.92			
	D6	1.082	0.029	37.429	***	0.92			
	D5	1.139	0.027	41.686	***	0.942			
	D4	1.142	0.031	37.137	***	0.928			
	D3	1.095	0.029	37.88	***	0.924			
	D2	0.973	0.034	28.996	***	0.860			
	D1	1.038	0.032	32.211	***	0.887			
Whistleblowing Behavior (E)	E1	1					0.756	0.916	0.978
	E2	1.198	0.033	36.808	***	0.747			
	E3	1.415	0.044	32.313	***	0.847			
	E4	1.293	0.045	28.438	***	0.948			
	E5	1.382	0.041	33.518	***	0.882			
	E6	1.152	0.046	25.062	***	0.965			
	E7	1.202	0.044	27.281	***	0.82			
Model Fit Index		*X*²(*p*) = 2158.801(0.000), *X*²/df = 4.050, RMSEA = 0.094, IFI = 0.924, CFI = 0.923, PGFI = 0.616, PNFI = 0.807							

***: *p* < 0.001.

**Table 4 behavsci-14-00293-t004:** The Results of Descriptive Statistics and Correlation Analysis.

	Mean	Standard Deviation	Organizational Ethical Climate	Organizational Identification	Person–Organization Value Congruence	Leader Ethical Behavior	Whistleblowing Behavior
Organizational Ethical Climate	5.908	0.986	-				
Organizational Identification	5.729	1.126	0.897 ***	-			
Person–Organization Value Congruence	5.686	1.145	0.862 ***	0.917 ***	-		
Leader Ethical Behavior	5.738	1.137	0.909 ***	0.899 ***	0.918 ***	-	
Whistleblowing Behavior	4.880	1.733	0.477 ***	0.569 ***	0.573 ***	0.548 ***	-

***: *p* < 0.001.

**Table 5 behavsci-14-00293-t005:** The results of Process Model 4.

Path			Estimate	S.E.	t	*p*	LLCI	ULCI
Organizational Ethical Climate	→	Organizational Identification	1.023	0.027	37.541	0.000	0.9703	1.0776
Organizational Ethical Climate	→	Whistleblowing Behavior	−0.300	0.176	−1.705	0.089	−0.6472	0.0461
Organizational Identification	→	Whistleblowing Behavior	1.112	0.154	7.206	0.000	0.8090	1.4164
Indirect effect(s) of X on Y								
Indirect Effect			Effect	Boot SE	Boot LLCI		Boot ULCI	
Organizational Ethical Climate → Organizational Identification → Whistleblowing Behavior			1.139	0.170	0.832		1.502	

**Table 6 behavsci-14-00293-t006:** The result of Moderation.

Dependent Variable: Organizational Identification							
	Model 1		Model 2		Model 3		
	*β*	*t*	*β*	*t*	*β*	*t*	VIF
Organizational Ethical Climate (A)	0.897 ***	37.542	0.413 ***	11.416	0.444 ***	11.385	4.575
Person–Organization Value Congruence (B)			0.561 ***	15.497	0.557 ***	15.441	3.911
Interaction					0.047 *	2.067	1.539
*R*^2^(Adjusted *R*^2^)	0.805(0.804)		0.885(0.885)		0.887(0.886)		
⊿*R*^2^(⊿Adjusted *R*^2^)	-		0.080(0.081)		0.002(0.001)		
F	1409.390 ***		1317.561 ***		888.224 ***		

***: *p* < 0.001, *: *p* < 0.05.

**Table 7 behavsci-14-00293-t007:** The moderated mediation effect of person–organization value congruence.

Dependent Variable: Whistleblowing Behavior					
Moderator	Level	Conditional Indirect Effect	Boot SE	Boot LLCI	Boot ULCI
Person–Organization Value Congruence	−1 SD (−1.1452)	0.5341	0.1270	0.3300	0.8062
	M	0.5642	0.1305	0.3586	0.8483
	+1 SD (1.1452)	0.5944	0.1364	0.3778	0.8937
Index of moderated mediation					
	Index		Boot SE	Boot LLCI	Boot ULCI
	0.0263		0.0155	0.0033	0.0644

**Table 8 behavsci-14-00293-t008:** The result of Moderation.

Dependent Variable: Organizational Identification							
	Model 1		Model 2		Model 3		
	*β*	*t*	*β*	*t*	*β*	*t*	VIF
Organizational Ethical Climate (A)	0.897 ***	37.542	0.460 ***	8.989	0.489 ***	9.231	6.232
Leader Ethical Behavior (B)			0.481 ***	9.410	0.485 ***	9.516	5.759
Interaction					0.054 *	2.006	1.592
*R*^2^(Adjusted *R*^2^)	0.805(0.804)		0.845(0.844)		0.847(0.845)		
⊿*R*^2^(⊿Adjusted *R*^2^)	-		0.040(0.040)		0.002(0.001)		
F	1409.390 ***		929.366 ***		626.414 ***		

***: *p* < 0.001, *: *p* < 0.05.

**Table 9 behavsci-14-00293-t009:** The result of Moderation.

Dependent Variable: Whistleblowing Behavior							
	Model 1		Model 2		Model 3		
	*β*	*t*	*β*	*t*	*β*	*t*	VIF
Organizational Ethical Climate (A)	0.477 ***	10.047	−0.121	−1.119	−0.063	−0.564	6.232
Leader Ethical Behavior (B)			0.659 ***	6.076	0.666 ***	6.159	5.759
Interaction					0.105 †	1.885	1.592
*R^2^*(Adjusted *R^2^*)	0.228(0.226)		0.303(0.299)		0.310(0.304)		
⊿*R^2^*(⊿Adjusted *R^2^*)	-		0.075(0.073)		0.007(0.005)		
F	100.947 ***		74.231 ***		50.989 ***		

***: *p* < 0.001, †: *p* < 0.1.

**Table 10 behavsci-14-00293-t010:** The moderated mediation effect of leader ethical behavior.

Dependent Variable: Whistleblowing Behavior					
Moderator	Level	Conditional Indirect Effect	Boot SE	Boot LLCI	Boot ULCI
Leader Ethical Behavior	−1 SD (−1.1379)	0.5883	0.1590	0.2997	0.9330
	M	0.6212	0.1600	0.3285	0.9660
	+1 SD (1.1379)	0.6541	0.1630	0.3586	1.0083
Index of moderated mediation					
	Index		Boot SE	Boot LLCI	Boot ULCI
	0.0289		0.0162	0.0066	0.0718

## Data Availability

The raw data supporting the conclusions of this article will be made available by the authors, without undue reservation.

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
