# Peer review of "Construed Organizational Ethical Climate and Whistleblowing Behavior: The Moderated Mediation Effect of Person–Organization Value Congruence and Ethical Leader Behavior"

_behavsci, 2024, doi:10.3390/bs14040293_

Round 1
Reviewer 1 Report
Comments and Suggestions for Authors
The authors presented a very interesting study about how an organization's ethical climate can affect employee's whistle blowing behavior. Despite the apparent effort you have put in, I do have a number of observations and concerns to share with you and hope you'd find them helpful.
1. Research motives: why this study is warranted? Your rational articulated in the front end basically says because these particular relationships had not been done before. There are many relationships have not been done, why these particular ones then?
2. The organization of the paper needs to be more clear and better aligned with conventional paper structure:
- Section 2.1 through 2.5: you spent a lot of pages to introduce each of your variables one by one and their relationships with each other. Then you moved on to build hypotheses. As a result, your writing is very repetitive and the logical flow is messy. More conventional way is you first introduce your main theories (1-1.5 pages). Then, you start to build the hypotheses. If any definitions needed, they could be presented with the hypothesis develop section.
- Your hypotheses: they are also overlapping with each other, and repetitive.
Mediation: the conventional structure is either you do X mediate relationships between IV and DV; Or, IV-M; then M-DV, and in this order. You will pick one instead of having both.
The same is true for moderated mediation relationships. I would recommend modeling after a couple quality published papers to restructure the hypotheses - basically, you don't need eight hypotheses - they are overlapping.
3. Organizational Ethical Climate:
- you did not provide an appropriate definition. On page 4, you basically defined a CLIMATE, not an ETHICAL climate.
- Your scale for ethical climate: Parboteeah's et al (2010) paper argued that there are three different type of organizational ethical climate instead of one as you seemed to imply. In addition, because you don't have a good definition for organizational climate, it is hard to judge which type fits your story. Regardless, combining items for all three types of ethical climates would not make sense whatsoever.
Comments on the Quality of English LanguageYour writing is very repetitive. The total length of the paper could be cut by one third at least. Moderate editing is needed.
Author Response
We strongly agree with the reviewer's comments and we have revised and supplemented according to the reviewer's comments.
Please see the attachment.

Reviewer 2 Report
Comments and Suggestions for Authors
Thank you for the opportunity to read this manuscript. The authors present a study on a current, relevant topic with strong theoretical and practical implications. The model is theoretically well-supported and the statistical analyses are complete. In order to ensure that the manuscript is published, I have just three suggestions for improvement:
- Halving the abstract.
- Revision of the hypothesis formulation. It's not clear what a "positive moderation" is. It would be clearer to explain the result of the model in the case of high/low levels of moderator presence.
- Given the complexity of the model and the number of hypotheses presented, I suggest introducing a final table and/or model with the systematization of the results (confirmation or non-confirmation) of the proposed hypotheses.
Thank you for the opportunity to read this manuscript and best wishes to the authors
Reviewer 3 Report
Comments and Suggestions for Authors
This is an interesting paper, and the topic of organizational culture is critical in understanding ethical behavior. I was surprised that the parallel topics of societal ethical culture and norms as well as the new literature on equity, inclusion and belonging were not brought in, as they clearly have a role in employee identification with an organization and their actions within the organization. There is also the topic of selection and hiring - is the organization expressing its cultural norms to prospective employees or asking employees to express their values and cultural expectations during the interview process? If there is corruption of some sort, and people understand that is how the company roles, then instances of that corruption may go unnoticed or acted upon. Trust can also protect and defend what others might consider unethical.
I really liked the straightforward presentation of the hypotheses and how that structure ran through the manuscript. That said I felt that the document, and in particular the introduction was unnecessarily wordy and repetitive. Also, some pieces seemed out of place, such as the statements on how "ethical leadership has a great impact on the behavior of an organization and individuals within it" on page 8. Yes, that is true, and seems fundamental; maybe move it up in the intro or skip restating it as I think that point is made earlier as well.
Regarding methods, again, you can scrub this section for unnecessary text. For example, the first sentence under same characteristics talks about what the study is looking at - readers should already be aware of this. you might provide a table for the demographic material, and understanding the response rate is important along with how this sample is representative of both the worker population of those offered to participate and the greater SME population. Lastly, do you have data on the number of whistle-blower complaints received for a particular population - for example the SME community, and extrapolate for the purpose of this study. Otherwise it is primarily opinions, and there is often response bias when talking about one's own ethical behavior.
for the analysis, it would be beneficial to provide the reader with a power analysis to understand if the p value was set properly. Also, there was a lot of detail in the front part of the section - explaining what a statistic is and how it works might be assumed rather than going into detail about it. It is also important how some of the results are stated - especially those that do not reach statistical significance. for example, the bottom of page 13 there is the phrase "...has not positive impact..." This might be reworded ....was not shown to have a positive impact." Also, at some points I felt the Results and Discussion sections were blurred. For example. page 18 "The results showed that...." - to me explaining what the results showed and mean is a discussion topic not a results topic.
With the discussion, I was surprised to see a focus on individualism v collectivism, which was not a theme in the paper up to this point. Also, I did expect to see more about organizational communications about its values (at different levels and through different means) and not so much about morals, which change. Going back to my prior comments, if people feel a sense of belonging and trust in the organization, that it is a fit with their own belief/value system then there may be more willingness to blow the whistle and highlight activities outside of those norms, rather than saying that there are absolute norms which people either fit or do not. I think it would also be helpful to refer back to the model in the discussion, highlighting connections that we now better understand and new work to be done. Under practical implications, it is unclear from what is written as to how to achieve a positive working climate - the most important elements that will impact it and one's willingness to blow the whistle. What are good measures of climate? level of presenteeism, turnover, engagement? It would also be interesting to hear your perspective as to how in sync the organization, leaders and employees need to be, and how consistent and persistent the communications and visible behaviors need to be to foster willingness to blow the whistle. Lastly, as previously mentioned, it would be helpful to understand the sample a bit better - especially its representativeness.
Comments on the Quality of English LanguageEnglish grammar and syntax generally fine, with some small errors. The larger concern is that there is a lot of duplication of material within and across sections - feels very robotic. The paper would be much clearer, and probably 1/4 shorter if there was not so much repetition. Pages 5 and 6 begin the repeating. My file does not have line numbers so I cannot be more specific. You also have incorrect hyphenation in places, for example page 7 lower half ("per-ceive" should be "perceive"). This is common through the manuscript.
Page 2: "organizational identification witness wrongdoing" is an awkward phrase and should be rewritten.
page 4 "rlated" should be related
page 5 run-on sentence starting" A strong..."
Page 5, link two sentence "...organizational identification, and encourages..."
Page 6: I think there is an extra word here "Ethical organizational ethical culture..."
Page 7: Extra word: "Therefore, if when the organization's..." Or say if or when.
Page 17: awkward/incomplete sentence: "To assess the conditional indirect...:
page 18: consider revision "Therefore, both hypothesis 4 is supported and through the results we can..."
Page 21: Consider revision "risks [8] in that employees curb..."
Page 21: "Employees in organizations..."
Page 21: Consider: "[1]. Also, employees with a high level of individualism may perceive their self-interests...with an ethical atmosphere, making them more reluctant to show whistleblowing behaviors [74]."
page 24: replace the word dissertation with study.
Round 2
Reviewer 1 Report
Comments and Suggestions for Authors
I appreciate authors' revision effort. The manuscript is much improved.
Comments on the Quality of English LanguageThe language is mostly OK.
